



Atmospheric
Chemistry
and Physics

Research article

# Pyruvic acid, an efficient catalyst in SO$_3$ hydrolysis and effective clustering agent in sulfuric-acid-based new particle formation

**Narcisse Tsona Tchinda**[1], **Lin Du**[1], **Ling Liu**[2], and **Xiuhui Zhang**[2]

[1]Environment Research Institute, Shandong University, Qingdao, 266237, China
[2]Key Laboratory of Cluster Science, Ministry of Education of China, School of Chemistry and Chemical Engineering, Beijing Institute of Technology, Beijing, 100081, China

**Correspondence:** Lin Du (lindu@sdu.edu.cn)

**Abstract.** The role of pyruvic acid (PA), one of the most abundant $\alpha$-keto carboxylic acids in the atmosphere, was investigated both in the SO$_3$ hydrolysis reaction to form sulfuric acid (SA) and in SA-based aerosol particle formation using quantum chemical calculations and a cluster dynamics model. We found that the PA-catalyzed SO$_3$ hydrolysis is a thermodynamically driven transformation process, proceeding with a negative Gibbs free-energy CE1 barrier, ca. $-1$ kcal mol$^{-1}$ at 298 K, $\sim 6.50$ kcal mol$^{-1}$ lower than that in the water-catalyzed SO$_3$ hydrolysis. Results indicated that the PA-catalyzed reaction can potentially compete with the water-catalyzed SO$_3$ reaction in SA production, especially in dry and polluted areas, where it is found to be $\sim 2$ orders of magnitude more efficient that the water-catalyzed reaction. Given the effective stabilization of the PA-catalyzed SO$_3$ hydrolysis product as SA·PA cluster, we proceeded to examine the PA clustering efficiency in a sulfuric-acid–pyruvic-acid–ammonia CE2 (SA-PA-NH$_3$) system. Our thermodynamic data used in the Atmospheric Cluster Dynamics Code indicated that under relevant tropospheric temperatures and concentrations of SA ($10^6$ cm$^3$ TS1), PA ($10^{10}$ cm$^3$) and NH$_3$ ($10^{11}$ and $5 \times 10^{11}$ cm$^3$), PA-enhanced particle formation involves clusters containing at most one PA molecule. Namely, under these monomer concentrations and 238 K, the (SA)$_2$·PA·(NH$_3$)$_2$ cluster was found to contribute by $\sim 100$ % to the net flux to aerosol particle formation. At higher temperatures (258 and 278 K), however, the net flux to the particle formation is dominated by pure SA-NH$_3$ clusters, while PA would rather evaporate from the clusters at high temperatures and not contribute to the particle formation. The enhancing effect of PA was examined by evaluating the ratio of the ternary SA-PA-NH$_3$ cluster formation rate to binary SA-NH$_3$ cluster formation rate. Our results show that while the enhancement factor of PA to the particle formation rate is almost insensitive to investigated temperatures and concentrations, it can be as high as $4.7 \times 10^2$ at 238 K and [NH$_3$] $= 1.3 \times 10^{11}$ molec. cm$^{-3}$. This indicates that PA may actively participate in aerosol formation, only in cold regions of the troposphere and highly NH$_3$-polluted environments. The inclusion of this mechanism in aerosol models may reduce uncertainties that prevail in modeling the aerosol impact on climate.

## 1   Introduction

Understanding the detailed processes involved in secondary aerosol formation continues to retain the attention of many researchers around the world. This is due to the varied role aerosols play in degrading visibility and human health, as well in affecting climate by altering cloud properties and influencing the balance of solar radiation (Stocker et al., 2013). Sulfuric acid (H$_2$SO$_4$, SA), which is believed to be the key species driving aerosol formation in the atmosphere (Kulmala et al., 2000; Kulmala, 2003; Sipila et al., 2010; Sihto et al., 2006; Kuang et al., 2008), is primarily formed from the hydrolysis of sulfur trioxide (SO$_3$). The main sources of atmospheric SO$_3$ include electrically neutral SO$_2$ oxidation by OH radicals and stabilized Criegee intermediates (Mauldin et al., 2012 `TS2`; Welz et al., 2012), whereas ion-induced oxidation constitutes a complementary source (Bork et al., 2013; Tsona et al., 2015a, 2016; Tsona and Du, 2019). The sinks of SO$_3$ include its reaction with water to produce sulfuric acid and acid rain, as well as reactions with other organic and inorganic species including ammonia and methanol (Li et al., 2018; Liu et al., 2019).

The general mechanism for SO$_3$ hydration to form sulfuric acid is a hydrogen atom transfer between H$_2$O and SO$_3$ within the SO$_3$·(H$_2$O)$_{n \geq 2}$ cluster, assisted by a second water molecule (Hofmann-Sievert and Castleman, 1984; Holland and Castleman, 1978), according to the following reaction:

$$SO_3 + nH_2O \rightarrow SO_3 \cdot (H_2O)_{n \geq 2} \rightarrow H_2SO_4 \cdot (H_2O)_{n-1}. \quad (R1)$$

In the presence of a single water molecule, the above reaction was found to be prevented by a high energy barrier, around 30 kcal mol$^{-1}$, whereas the barrier height gradually decreases to ca. ∼ 0 kcal mol$^{-1}$ as the number of water molecules increases to 4 or more (Hofmann and Schleyer, 1994; Morokuma and Muguruma, 1994; Larson et al., 2000; Loerting and Liedl, 2000). In Reaction (R1), the second water molecule acting as a catalyst forms a bridge for the hydrogen atom to transfer from H$_2$O to SO$_3$. This reaction is considered as the major loss pathway for SO$_3$ in the atmosphere. Beside a second water molecule, a number of studies have shown that the SO$_3$ + H$_2$O → H$_2$SO$_4$ reaction can be facilitated by organic and inorganic species including sulfuric acid, formic acid, nitric acid and oxalic acid (Torrent-Sucarrat et al., 2012; Hazra and Sinha, 2011; Daub et al., 2020; Long et al., 2013; Lv et al., 2019). In the presence of these species, the SO$_3$ + H$_2$O reaction can effectively proceed in a near barrierless mechanism, where they also act as bridge for hydrogen atom transfer. However, due to the low concentration of each of these catalysts, their overall effect on the rate of sulfuric acid formation from the SO$_3$ + H$_2$O reaction is not strong enough that they cannot effectively compete with the water-catalyzed reaction, given the relatively high concentration of water. An efficient catalyst would not only promote a fast hydrogen transfer between SO$_3$ and H$_2$O,

but would also have a high enough concentration to induce a net higher reaction rate than the water.

Pyruvic acid (CH$_3$C(O)COOH, PA), the simplest and one of the most abundant $\alpha$-keto acids in the troposphere, is highly present in plants and in tropospheric air (Eisenreich et al., 2001; Jardine et al., 2010; Magel et al., 2006; Eger et al., 2020). Sources of PA in tropospheric air include photooxidation of isoprene, photolysis of methyglyoxal, reactions of peroxy radicals formed during the oxidation of propane and the photooxidation of aromatic compounds in the presence of NO$_x$, as well as vegetation (Paulot et al., 2009; Jenkin et al., 1993; Warneck, 2005; Praplan et al., 2014; Talbot et al., 1990; Jardine et al., 2010). PA mixing ratios of up to 15, 25 and 96 ppt were reported in the free troposphere and forest canopy over central Amazonia, and in a Finnish boreal forest, respectively (Talbot et al., 1990; Eger et al., 2020). In a rural continental mountain top site over the eastern US, PA mixing ratios reaching 200 ppt were measured (Talbot et al., 1995), while levels as high as 800 ppt were found above the equatorial African rainforest (Helas et al., 1992). In regions highly affected by anthropogenic activities, PA levels of up to 500 ppt were observed, whereas significantly low levels were observed in the marine boundary layer over the Atlantic Ocean (63° N to 39° S) (Baboukas et al., 2000; Mattila et al., 2018). PA is expected to be removed from the atmosphere through photolysis and oxidation by OH radicals (Mellouki and Mu, 2003; Reed Harris et al., 2016, 2017a, b; Church et al., 2020).

The detection of PA in various media, including gas-phase, aerosol and aqueous-phase (Andreae et al., 1987; Talbot et al., 1990; Bardouki et al., 2003; Baboukas et al., 2000; Chebbi and Carlier, 1996; Kawamura et al., 1996, 2013; Kawamura and Bikkina, 2016), makes it a good candidate for atmospheric processes. Moreover, through its carboxyl and carbonyl functions, PA can partake not only in hydrogen atom transfer reactions but also in molecular clustering owing to its ability to form hydrogen bonds. In this study, we examine the catalytic effect of PA on the SO$_3$ hydration to form SA and assess the subsequent clustering of the reaction products (PA·SA) with additional SA and ammonia molecules. The kinetics of the SO$_3$ hydrolysis are determined, and the fate of the product cluster in atmospheric particle formation is evaluated from cluster dynamic simulations.

## 2   Methodology

### 2.1   Quantum chemical calculations

The calculations in this study were divided into reaction mechanism and cluster formation parts, and all geometry optimizations were performed using the Gaussian 09 package (Frisch et al., 2013). In the reaction mechanism part, the configurations of different states of the scanned reaction pathways were initially optimized with the M06-2X density functional (Zhao and Truhlar, 2008) used in conjunction with

the 6-31+G(d,p) basis set. Identified M06-2X/6-31+G(d,p) structures within $3 \, \text{kcal} \, \text{mol}^{-1}$ of the lowest energy structure were re-optimized, followed by vibrational frequency analysis at the M06-2X/6-311++G(3df,3pd) level of theory, thereby yielding zero-point energies as well as thermal correction to Gibbs free energies. It should be noted that vibrational frequency calculations were performed under the harmonic oscillator and rigid rotor approximation at 298 K and 1 atm. Transition state configurations were determined using the synchronous transit quasi-Newton method (Peng et al., 1996) and confirmed by intrinsic reaction coordinate calculations (Fukui, 1981) to ensure they connected the reactants to desired products. Electronic energies of all M06-2X/6-311++G(3df,3pd) optimized structures were corrected by the DLPNO-CCSD(T)/aug-cc-pVTZ method using Orca version 4.2.1 (Riplinger et al., 2013; Riplinger and Neese, 2013).

## 2.2  Kinetics

The kinetic analysis of the studied reactions was performed following the conventional transition state theory (Duchovic et al., 1996; Truhlar et al., 1996) with the Wigner tunneling correction and executed with the KiSThelP program (Canneaux et al., 2014). Starting from the initial reactants (SO₃, H₂O and PA), the different processes in these reactions include the collision of two separate reactants to form a binary complex that further interact with the third species to form the pre-reactive intermediate, the evaporation of the pre-reactive intermediate to initial reactants and its forward reaction to form the products, according to the following reaction:

$$\text{SO}_3 + \text{H}_2\text{O} \leftrightarrow \text{SO}_3 \cdots \text{H}_2\text{O}, \tag{R2a}$$

$$\text{SO}_3 \cdots \text{H}_2\text{O} + X \leftrightarrow \quad \text{Pre-reactive intermediate}$$
$$\rightarrow \text{H}_2\text{SO}_4 \cdots X \rightarrow \text{H}_2\text{SO}_4 + X. \tag{R2b}$$

Reactions (R2a) and (R2b) are representative examples of processes taking place in the SO₃ hydrolysis with $X$ as catalyst, where $X$ is H₂O or PA. It should be noted that the pre-reactive intermediate can also form from the formation of the binary complex between the catalyst and water, followed by its interaction with SO₃. Recent studies showed that no matter the order in which the binary complex and the pre-reactive intermediate are formed and regardless of the specific reactants involved in the formation of the binary complex, the interactions between SO₃, H₂O and the catalyst lead to the formation of sulfuric acid plus catalyst (Weber et al., 2001; Jayne et al., 1997; Torrent-Sucarrat et al., 2012; Hazra and Sinha, 2011; Lv et al., 2019). Assuming equilibrium between the reactants and the complex, and steady-state approximation of the pre-reactive intermediate, the overall rate of Reaction (R2) with catalyst $X$ is

$$\begin{aligned} v_x &= \frac{k_1}{k_{-1}} \frac{k_2}{k_{-2}} k_{\text{uni},x} [\text{SO}_3][\text{H}_2\text{O}][X] \\ &= K_{\text{eq1}} K_{\text{eq2}} k_{\text{uni},x} [\text{SO}_3][\text{H}_2\text{O}][X] \\ &= k_x [\text{SO}_3][\text{H}_2\text{O}][X], \end{aligned} \tag{1}$$

where $k_1$ is the collision frequency of SO₃ and H₂O to form the SO₃⋯H₂O binary complex, and $k_{-1}$ is the evaporation rate constant of SO₃⋯H₂O back to initial reactants, $k_2$ is the collision frequency of SO₃⋯H₂O and the catalyst to form the pre-reactive intermediate, $k_{-2}$ is the evaporation rate constant of the pre-reactive intermediate back to its precursor reactants. $K_{\text{eq1}}$ and $K_{\text{eq2}}$ are equilibrium constants of formation of the binary complex and pre-reactive intermediate, respectively; $k_{\text{uni},x}$ is the unimolecular rate constant of the reaction of the pre-reactive intermediate to the products; and $k_x$ is the overall rate constant of Reaction (R2) in the presence of catalyst $X$. [SO₃], [H₂O] and $[X]$ are respective gas-phase concentrations of SO₃, H₂O and the catalyst (H₂O or PA). The determination of equilibrium constants and unimolecular rate constants was executed using the KiSThelP program, and their numerical values are given in Tables S2 and S3 in the Supplement.

## 2.3  Cluster formation and dynamic simulations

As Reaction (R2) results in the formation of sulfuric acid complexed to the catalyst, we explored the thermodynamics of further clustering with more PA, sulfuric acid (SA) and ammonia (NH₃) molecules. Although sulfuric acid as a monomer is known to cluster effectively with water at relevant atmospheric conditions, several studies have demonstrated that the binding strength of water significantly decreases as sulfuric acid is clustered to other sulfuric acid and base molecules (Temelso et al., 2012a, b; Henschel et al., 2014; Tsona et al., 2015b). Moreover, cluster dynamics simulations have shown that despite its clustering to sulfuric acid and sulfuric-acid-based clusters, water rapidly evaporates from the cluster owing to its high evaporation rate, particularly when base molecules are present in the cluster. As a result, most atmospheric sulfuric-acid-based clusters are detected in their dry state, exclusively (Almeida et al., 2013; Olenius et al., 2013b). It follows that though water would play a certain role in the cluster thermodynamics, its net effect on particle formation rate is negligible. Hence, water was not included in the studied clusters.

A number of initial configurations of SA–NH₃ clusters were taken from previously published results (Ortega et al., 2012) and re-optimized with the M06-2X/6-31++G(d,p), while those containing PA were built by the stepwise addition of monomers to the relevant cluster. On this basis, several starting configurations were generated manually by arranging the participating molecules or clusters in different directions. Depending on the cluster size, 10–30 initial configu-

rations of each cluster were pre-optimized at the M062X/6-31+G(d) level of theory. All identified structures within 3 kcal mol$^{-1}$ of the lowest energy structure were thereafter re-optimized with the M06-2X/6-31++G(d,p) method, and the vibrational frequency analysis was subsequently performed at the same level of theory. It has been shown that the reduction from the 6-311++G(3df,3pd) to 6-31++G(d,p) basis set for sulfuric-acid-based cluster formation induces very little error in the thermal contribution to the Gibbs free energy, with no further substantial effect on the single-point energy, yet sufficiently reducing the computation cost (Elm and Mikkelsen, 2014). The electronic energies of M06-2X/6-31++G(d,p) optimized structures were further corrected with the DLPNO-CCSD(T)/aug-cc-pVTZ method. To elucidate the role of PA in atmospheric particle formation, the influence of varying temperatures and monomer concentrations on SA-PA-NH$_3$ clustering was examined using the Atmospheric Cluster Dynamics Code (ACDC; McGrath et al., 2012). The model cluster is $(SA)_s\cdot(PA)_p\cdot(NH_3)_n$ ($0 \leq n \leq s+p \leq 3$). The simulation box was set to $3 \times 2$, where "3" stands for the total number of acids (SA and PA) and "2" stands for the total number of NH$_3$ molecules. The ACDC model, taking as input the thermodynamic data obtained from quantum chemical calculations, generates the time derivatives of all cluster concentrations and solves for the steady-state cluster distribution, using the MATLAB ode 15s routine for differential equations (Shampine and Reichelt, 1997). The time derivatives of cluster concentrations, also called birth–death equations, can be expressed as follows:

$$\frac{dC_i}{dt} = \frac{1}{2}\sum_{j<i}\beta_{j,(i-j)}C_jC_{i-j}$$
$$+ \sum_j\gamma_{(i+j)\rightarrow i,j}C_{i+j} - \sum_j\beta_{i,j}C_iC_j$$
$$- \frac{1}{2}\sum_{j<i}\gamma_{i\rightarrow j,(i-j)}C_i + Q_i - S_i, \quad (2)$$

where $C_i$ is the concentration of cluster $i$, $\beta_{i,j}$ is the collision coefficient of clusters $i$ and $j$, and $\gamma_{k\rightarrow i,j}$ is the rate coefficient of cluster $k$ evaporating into smaller clusters $i$ and $j$. $Q_i$ and $S_i$ are the possible outside source term and sink term, respectively, for cluster $i$.

For two neutral clusters $i$ and $j$, the collision coefficient under the assumption of hard-sphere and sticking collision was calculated according to the kinetic gas theory as

$$\beta_{i,j} = \left(\frac{3}{4\pi}\right)^{1/6}\left[6k_BT\left(\frac{1}{m_i} + \frac{1}{m_j}\right)\right]^{1/2}\left(V_i^{1/3} + V_j^{1/3}\right)^2, \quad (3)$$

where $m_i$ and $V_i$ are the respective mass and volume of cluster $i$, $k_B$ is the Boltzmann constant and $T$ is the absolute temperature. The volume is calculated from atomic masses and densities of the compounds in the cluster.

The rate coefficient of the $i+j$ cluster evaporating to $i$ and $j$ clusters was derived as

$$\gamma_{(i+j)\rightarrow i,j} = \beta_{i,j}\frac{P_{ref}}{k_BT}\exp\left(\frac{\Delta G_{i+j} - \Delta G_i - \Delta G_j}{k_BT}\right), \quad (4)$$

where $P_{ref}$ is the reference pressure at which Gibbs free energies are calculated, $\Delta G_i$ is the Gibbs free energy of formation of cluster $i$ from monomers. Further details on collision rate coefficients and evaporation rate coefficients evaluation as well as on ACDC simulations can be obtained from previous studies (McGrath et al., 2012; Ortega et al., 2012; Olenius et al., 2013a, b).

## 3 Results and discussion

### 3.1 Water-catalyzed SO$_3$ hydrolysis

A number of studies have been dedicated to the $SO_3 + H_2O \rightarrow H_2SO_4$ reaction, which was shown to be prevented by an electronic energy barrier as high as $\sim 30$ kcal mol$^{-1}$ under relevant atmospheric conditions. The presence of a second H$_2$O molecule in this reaction lowers the energy barrier by favoring the formation of two kinds of binary hydrogen-bonded complexes, $SO_3\cdots H_2O$ and $H_2O\cdots H_2O$, which can interact thereafter with the third species, H$_2$O and SO$_3$, respectively, to form a ternary pre-reactive intermediate, $SO_3\cdots H_2O\cdots H_2O$. Indeed, due to the difficulty of termolecular interactions relative to biomolecular interactions (Buszek et al., 2012), the most likely situation is the interaction of two species to form a two-body complex followed by the interaction with the third species to form the pre-reactive intermediate. As shown on the energy surface in Fig. 1, though the formation of $SO_3\cdots H_2O$ is slightly more favorable than $H_2O\cdots H_2O$ formation, both complexes form $SO_3\cdots H_2O\cdots H_2O$, henceforth denoted RC, that lies at 8.73 kcal mol$^{-1}$ electronic energy below $SO_3\cdots H_2O$ and 12.50 kcal mol$^{-1}$ below $H_2O\cdots H_2O$. These energies are respectively within 0.17–1.81 and 0.47–1.06 kcal mol$^{-1}$, similar to previously reported values for the same reaction (Hazra and Sinha, 2011; Torrent-Sucarrat et al., 2012; Long et al., 2013; Lv et al., 2019). The slight observed differences likely result from the differences in the computational approaches used in these studies. This intermediate overcomes an electronic energy barrier of 5.47 kcal mol$^{-1}$ to form $H_2SO_4\cdots H_2O$. This energy barrier is $\sim 25$ kcal mol$^{-1}$ lower than the barrier in the reaction without the second water molecule and in good agreement with previous results (Torrent-Sucarrat et al., 2012; Hazra and Sinha, 2011; Long et al., 2013; Lv et al., 2019). The underlying mechanism is similar to those previously reported for the same reaction and is characterized by two hydrogen atom transfers to and from the second water molecule that acts as the catalyst.

The unimolecular decomposition of $SO_3\cdots H_2O\cdots H_2O$ to form $H_2SO_4\cdots H_2O$ occurs at a rate constant of

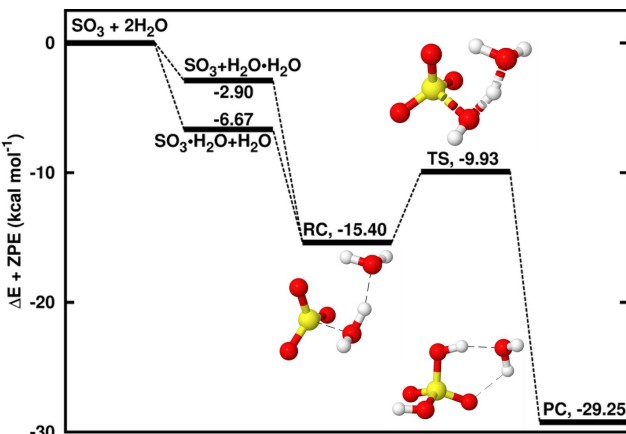

**Figure 1.** Energy surface for the water-catalyzed SO$_3$ hydrolysis. "RC" stands for pre-reactive intermediate; "TS" is transition state and "PC" is product complex. Color coding is yellow for sulfur atom, red for oxygen atom and white for hydrogen atom. Electronic energy values of all intermediates are indicated, and corresponding Gibbs free values are given in Table S4 in the Supplement.

$1.35 \times 10^8$ s$^{-1}$ at 298 K and, taking into account the collision and evaporation processes driving the formation of the binary complexes and SO$_3 \cdots$H$_2$O$\cdots$H$_2$O, the overall rate constant of the water-catalyzed SO$_3$ hydrolysis at 298 K is determined to be $1.09 \times 10^{-32}$ cm$^6$ molec.$^{-2}$ s$^{-1}$.

### 3.2 Pyruvic acid-catalyzed SO$_3$ hydrolysis

Despite the demonstrated catalytic effect of water on SO$_3$ hydrolysis, comparison of previous studies demonstrates that as many as four water molecules could be needed to achieve similar results to those obtained when a single molecule of other species is used as catalyst (Hazra and Sinha, 2011; Torrent-Sucarrat et al., 2012; Long et al., 2013; Lv et al., 2019; Daub et al., 2020; Larson et al., 2000). Some of these species include carboxylic acids, sulfuric acid and nitric acid. From these studies, while the electronic energy barrier could be reduced to $\sim 5.5$ kcal mol$^{-1}$ with water as the catalyst, it could be reduced to 3.7, 1.4, 0.6 and around 1 kcal mol$^{-1}$, respectively, with HNO$_3$, H$_2$SO$_4$, HCOOH and HOOC-COOH as catalysts. These results highlight the catalytic strength of carboxylic acids over other catalysts and further suggest that a second carboxyl function can have additional catalytic effects on the energy barrier.

In addition to the carboxyl group, PA possesses a ketone function at the $\alpha$ position. Church et al. have identified four stable conformational structures for PA that mainly differ by the orientation of the methyl group relative to the acidic OH group and that of the hydroxyl H atom relative to the ketone group, leading to *trans-cis* (Tc), *trans-trans* (Tt), *cis-trans* (Ct) and *cis-cis* (Cc) conformers, denoted as PA$_{Tc}$, PA$_{Tt}$, PA$_{Ct}$ and PA$_{Cc}$ conformers, respectively. These structures are shown in Fig. S1 in the Supplement, along with

their energies given relative to the energy of the most stable conformer, PA$_{Tc}$ (Church et al., 2020). Only three of these PA conformations were able to form complexes with water: PA$_{Tc}$, PA$_{Tt}$ and PA$_{Ct}$. We also note that during geometry optimization, PA$_{Tc}$ could interact with H$_2$O and be converted into PA$_{Tt}$, forming the binary PA$_{Tt}\cdots$H$_2$O complex. Henceforth, the PA$_{Tt}\cdots$H$_2$O will represent the complex resulting from PA$_{Tc}$ + H$_2$O and PA$_{Tt}$ + H$_2$O interactions.

Similar to the SO$_3$ hydrolysis where water acts as the catalyst, the reaction with PA acting as catalyst proceeds by formation of SO$_3\cdots$H$_2$O and PA$\cdots$H$_2$O complexes prior to the formation of the PA$\cdots$H$_2$O$\cdots$SO$_3$ pre-reactive intermediate (see Fig. 2). While the pre-reactive intermediates formed with PA$_{Tt}$ and PA$_{Ct}$ conformers have almost equal electronic energy of formation, the PA$_{Tt}$ conformer is 0.23 kcal mol$^{-1}$ more stable than the PA$_{Ct}$ conformer with respect to the Gibbs free energy at ambient conditions. Regardless of the PA conformation, the transformation of the pre-reactive intermediate to form PA$\cdots$SA follows hydrogen transfer mechanisms where the hydrogen atom transfers from water to PA and then from PA to SO$_3$, releasing the PA$\cdots$SA complex, similar to the mechanisms where water, formic acid and oxalic acid act as catalysts. PA$_{Tt}$ and PA$_{Ct}$ conformers exhibit comparable binding strength with water at 1 atm and 298 K, with 2.75 and 2.29 kcal mol$^{-1}$ Gibbs free-energy changes, respectively. This is a somewhat more favorable binding than the SO$_3$ binding with water whose Gibbs free-energy change is 3.37 kcal mol$^{-1}$ at similar conditions. It is obvious that the formation of PA$\cdots$H$_2$O$\cdots$SO$_3$ pre-reactive intermediate would preferably proceed through PA$\cdots$H$_2$O + SO$_3$ rather than PA + H$_2$O$\cdots$SO$_3$, although both interactions would lead to the same pre-reactive intermediate. Moreover, to evaluate the impact of each of these paths to the formation of the pre-reactive intermediate, we determined that among the two hydrates, PA$\cdots$H$_2$O will contribute by more than 99.99 % while SO$_3\cdots$H$_2$O will contribute by less than 0.01 % to the formation of the pre-reactive intermediate regardless of the PA conformer (details are given in the Supplement).

The PA$_{Tt}\cdots$H$_2$O$\cdots$SO$_3$ pre-reactive intermediate is formed with 0.84 and 0.24 kcal mol$^{-1}$ Gibbs free-energy changes relative to PA$_{Tt}\cdots$H$_2$O + SO$_3$ and PA$_{Tt}$ + SO$_3\cdots$H$_2$O interactions, respectively. While there is an electronic energy barrier as low as 0.45 kcal mol$^{-1}$ separating PA$_{Tt}\cdots$H$_2$O$\cdots$SO$_3$ from the product, this transformation is rather thermodynamically driven, with a negative Gibbs free-energy barrier of $-0.92$ kcal mol$^{-1}$ and the product complex lying at 13.30 Gibbs free energy below PA$_{Tt}\cdots$H$_2$O + SO$_3$. Note that in the transformation of PA$_{Tt}\cdots$H$_2$O$\cdots$SO$_3$ to the product complex (PC$_{Tt}$), the PA$_{Tt}$ conformer is isomerized to PA$_{Ct}$. This phenomenon has also been observed in SO$_3$ hydrolysis reactions catalyzed by oxalic acid (Lv et al., 2019).

The PA$_{Ct}\cdots$H$_2$O$\cdots$SO$_3$ pre-reactive intermediate is formed with 0.20 and $-0.86$ kcal mol$^{-1}$ Gibbs free-energy changes relative to PA$_{Ct}\cdots$H$_2$O + SO$_3$ and

https://doi.org/10.5194/acp-22-1-2022

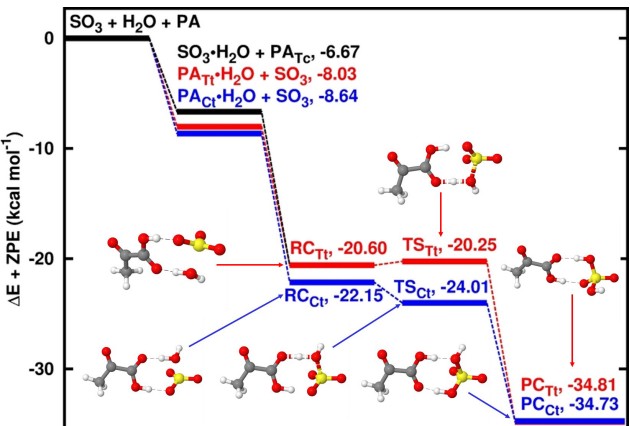

**Figure 2.** Energy surfaces for the pyruvic-acid-catalyzed CE5 $SO_3$ hydrolysis. "RC" stands for pre-reactive intermediate; "TS" is transition state and "PC" is product complex. Color coding is yellow for sulfur atom, red for oxygen atom, grey for carbon atom and white for hydrogen atom. Colored lines indicate the different paths involving the *trans-trans* and *cis-trans* conformers of pyruvic acid. Electronic energy values of all intermediates are indicated, and corresponding Gibbs free values are given in Table S4 in the Supplement.

$PA_{Ct} + SO_3 \cdots H_2O$ interactions, respectively, and is separated from the product complex by a $0.09\,\mathrm{kcal\,mol^{-1}}$ electronic energy barrier. The negative Gibbs free-energy barrier of $-1.15\,\mathrm{kcal\,mol^{-1}}$ in this path indicates the thermodynamically driven transformation process. This path is somewhat more favorable than the path catalyzed by $PA_{Tt}$ at atmospheric pressure and 298 K. Similar to the path catalyzed by $PA_{Tt}$, $PA_{Ct}$ is isomerized to $PA_{Tt}$ during the $PA_{Ct} \cdots H_2O \cdots SO_3 \rightarrow PC_{Ct}$ transformation process.

Based on the transition state energies, the unimolecular decompositions of $PA_{Tt} \cdots H_2O \cdots SO_3$ and $PA_{Ct} \cdots H_2O \cdots SO_3$ pre-reactive intermediates were determined to be $9.03 \times 10^{11}$ and $1.80 \times 10^{12}\,\mathrm{s^{-1}}$, respectively, at 298 K. Corresponding overall rate constants for the PA-catalyzed $SO_3 + H_2O$ reaction, calculated according to Eq. (1), are $2.95 \times 10^{-27}$ and $3.52 \times 10^{-26}\,\mathrm{cm^6\,molec.^{-2}\,s^{-1}}$, respectively. Considering the Gibbs free energies of $PA_{Tc}$, $PA_{Tt}$ and $PA_{Ct}$, and applying the law of mass action on the equilibria between $PA_{Tc}$, $PA_{Tt}$ and $PA_{Ct}$, the determined relative equilibrium distribution of PA conformers is 0.95 ($PA_{Tc}$), 0.04 ($PA_{Tt}$) and 0.01 ($PA_{Ct}$) at 298 K (Table S1 in the Supplement). Considering this distribution, the weighted average rate constant of PA-catalyzed $SO_3$ hydrolysis at 298 K is determined to be $3.10 \times 10^{-27}\,\mathrm{cm^6\,molec.^{-2}\,s^{-1}}$, being $2.84 \times 10^5$ times higher than the rate constant of the water-catalyzed $SO_3$ hydrolysis at 298 K. This ratio is much higher than observed with other catalysts, for example, 1.19 with nitric acid (Long et al., 2013), $10^2$ with sulfuric acid (Torrent-Sucarrat et al., 2012), $10^3$ with oxalic acid (Lv et al., 2019) and $10^4$ with formic acid (Hazra and Sinha, 2011). However, given the low

atmospheric concentrations of these species relative to water, the effective rate of the $SO_3$ hydrolysis reactions where they act as catalysts are not high enough for them to effectively compete with the water-catalyzed reaction. For a reaction to compete with the water-catalyzed $SO_3$ hydrolysis, not only should the catalyst be efficient in facilitating the hydrogen transfer between $H_2O$ and $SO_3$, but its concentration must be high enough to cause a higher SA formation rate than the water-catalyzed reaction.

Comparing the rate constant of the PA-catalyzed $SO_3$ hydrolysis reaction to the rate constants of the reaction catalyzed by other species mentioned above, it is obvious that PA is the most efficient catalyst. Consequently, PA may be an efficient partaker in $SO_3$ hydrolysis in the atmosphere. To effectively compare the different catalytic effects of PA and water on the $SO_3 + H_2O$ reaction, it is important to compare the overall rates of SA formation (given in Eq. 1) that take into account the concentrations of the catalysts. From Eq. (1), the ratio of the rates of $SO_3$ hydrolysis reactions catalyzed by PA and water can be expressed as

$$\alpha = \frac{k_{PA}}{k_{H_2O}} \times \frac{[PA]}{[H_2O]}, \tag{5}$$

where $\alpha$ is a measure of the relative efficiency of different catalysts and $k_x$ is the overall rate constant of the $X$-catalyzed $SO_3$ hydrolysis ($X = $ PA, $H_2O$), given in Eq. (1). Assuming water concentrations within $10^{15}$–$10^{17}\,\mathrm{molec.\,cm^{-3}}$ that cover dry and humid conditions, and PA concentrations in the range $10^9$–$10^{11}\,\mathrm{cm^{-3}}$ TS3 that cover clean and polluted environments, our results show that the efficiency of PA as a catalyst lies in the range $\sim 10^{-2}$–$10^2$ relative to water. The $H_2O$-catalyzed $SO_3$ hydrolysis remains the main $SO_3$ loss pathway under humid conditions, whereas the PA-catalyzed $SO_3$ hydrolysis would be the dominant path under dry conditions and in polluted areas where PA concentrations can reach the ppb levels. It is estimated that under such conditions, the PA-catalyzed reaction can be around 2 orders of magnitude more efficient than the $H_2O$-catalyzed $SO_3$ hydrolysis to form sulfuric acid. This shows, in regard to the relatively high PA concentration and the high rate constant of the PA-catalyzed $SO_3$ hydrolysis, that this reaction is more effective for SA production than reactions catalyzed by formic acid, sulfuric acid and oxalic acid. Given the renowned role of sulfuric acid in aerosol particle formation, the PA-catalyzed $SO_3$ hydrolysis, stabilized as the PA$\cdots$SA complex product, might provide an additional pathway for incorporating organic matter into aerosol particles.

## 3.3   Cluster thermodynamics and dynamic simulations

### 3.3.1   Cluster thermodynamics

The thermodynamics of further SA-PA clustering, with and without ammonia ($NH_3$), was examined. In general, the cluster formation is thermodynamically favorable at various tropospheric temperatures as can be seen in Table S5 in

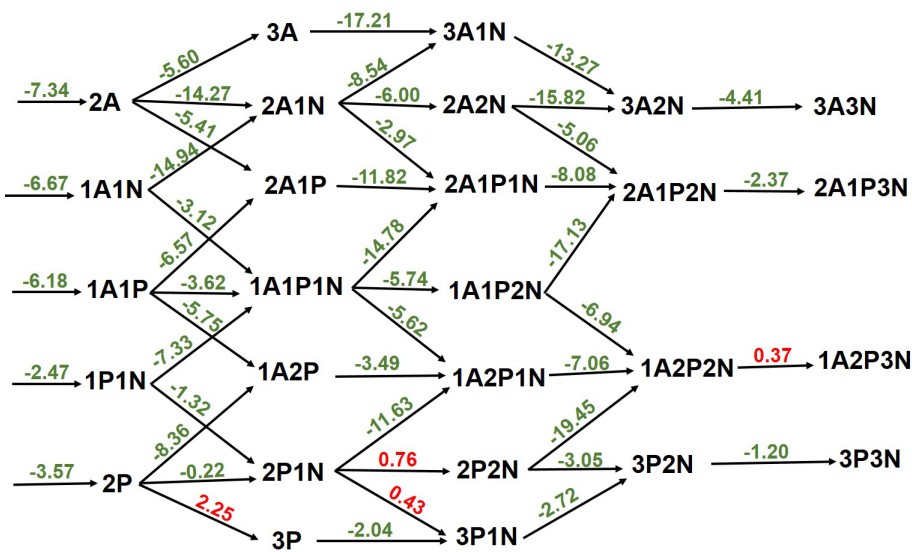

**Figure 3.** Gibbs free-energy (in kcal mol⁻¹) diagram of sulfuric-acid–pyruvic-acid–ammonia clusters at 298 K and 1 atm. "A" refers to sulfuric acid, "P" refers to pyruvic acid and "N" refers to ammonia. Green color indicates exergonic processes and red color indicates endergonic processes.

the Supplement. The binding of PA to SA exhibits similar strength within 1 kcal mol⁻¹ to the binding between two SA molecules, though this binding is weakened in the presence of NH₃, likely as a result of the weaker acid nature of PA than SA. Figure 3 exhibits the free-energy diagram for cluster formation steps in the SA-PA-NH₃ system at 298 K and 1 atm, and it shows that the stepwise formation of the studied clusters is in general exergonic, with only four moderate endergonic processes involving PA and NH₃ additions. According to our thermodynamic data, PA additions to (PA)₂ and (PA)₂·NH₃ are hindered by thermodynamic barriers of 2.25 and 0.43 kcal mol⁻¹, respectively, while NH₃ additions to (PA)₂·NH₃ and SA·(PA)₂·(NH₃)₂ are hindered by 0.76 and 0.37 kcal mol⁻¹ thermodynamic barriers, respectively. Similar additions to clusters containing SA instead of PA are much more exergonic, and this is as expected given the strong binding between SA and NH₃, as compared with the binding between PA and NH₃. The cluster formation depicted in the energy diagram of Fig. 3 is based on direct quantum chemical data at 298 K and 1 atm and does not take into account the actual concentrations of monomers. However, such processes in the real atmosphere depend on actual atmospheric conditions such as temperature and the concentrations of monomers participating in the process

### 3.3.2  Steady-state cluster concentrations and cluster formation rates

Time-dependent cluster concentrations were determined by solving the birth–death equations given in Eq. (2), using ACDC, under given concentrations of monomers. We found that under [SA] = 10⁶–

10⁸ cm⁻³ TS4, [PA] = 10⁷–10¹⁰ cm⁻³ TS5 and [NH₃] = 10⁸–10¹¹ cm⁻³ conditions, the largest PA-containing cluster is (SA)₂·PA·(NH₃)₂, with the highest concentration observed under 238 K, [SA] = 10⁶ cm⁻³, [PA] = 10¹⁰ cm⁻³ and [NH₃] = 10¹¹ cm⁻³ conditions. This concentration is, however, still as low as ~ 10 cm⁻³. Cluster equilibrium concentrations at selected representative conditions are shown in Fig. 4. Previous studies also found that in sulfuric-acid-based particle formation enhanced by methane sulfonic acid (MSA) and trifluoroacetic acid (TFA), the main cluster contributing to particle formation would bind a single molecule of MSA or TFA (Bork et al., 2014; Lu et al., 2020). This is due to the particular binding strength between the base molecule and SA, as compared to the binding with other acids.

The difficulty to form large clusters containing PA likely result from the low concentration of heterotrimers and tetramers containing PA. For example, it is seen from Fig. 4 that (SA)₂·PA, SA·PA·NH₃, SA·PA·(NH₃)₂ and (SA)₂·PA·(NH₃) all have steady-state concentrations below 10 cm⁻³, although the concentrations of heterodimers containing PA (SA·PA and PA·NH₃) can reach 10⁵ cm⁻³. These heterodimers likely undergo faster evaporation than, for example, SA·SA and SA·NH₃, hence not effectively contributing to further growth. The highest concentrations of PA-containing clusters are found at low temperatures, exclusively, where the evaporation rates are reduced.

The contribution of PA to the particle formation was estimated by calculating the enhancement factor as

$$r = \frac{J\left([SA] = 10^6, [PA] = x, [NH_3 = y]\right)}{J\left([SA] = 10^6, [PA] = 0, [NH_3 = y]\right)}, \quad (6)$$

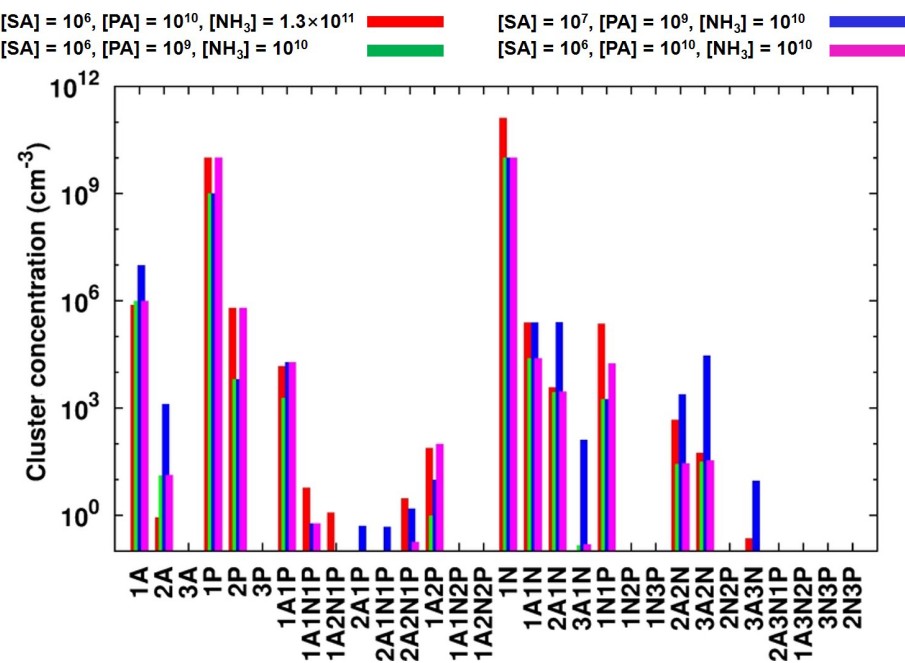

**Figure 4.** Simulated steady-state clusters distribution in the sulfuric-acid–pyruvic-acid–ammonia system at 238 K, under different initial monomers concentrations. "A" refers to sulfuric acid, "P" refers to pyruvic acid and "N" refers to ammonia.

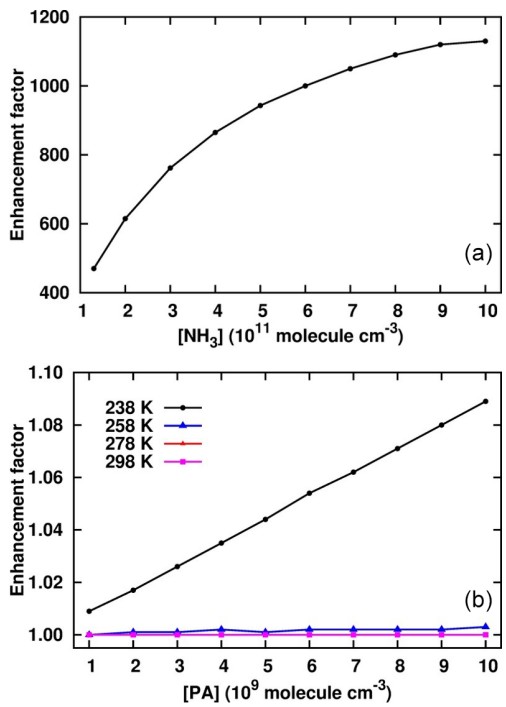

**Figure 5.** Enhancement of PA in the cluster formation rate in the sulfuric-acid–pyruvic-acid–ammonia clusters at $[SA] = 10^6 \text{ cm}^{-3}$, $[NH_3] = 10^{10} \text{ cm}^{-3}$, $[PA] = 10^9\text{–}10^{10} \text{ cm}^{-3}$ and different temperatures **(b)**, and $T = 238$ K, $[SA] = 10^6 \text{ cm}^{-3}$, $[PA] = 10^{10} \text{ cm}^{-3}$, $[NH_3] = 10^{11}\text{–}10^{12} \text{ cm}^{-3}$ TS6 **(a)**.

where $J$ is the cluster formation rate calculated at fixed SA concentration ($10^6 \text{ cm}^{-3}$ TS7), $x = 10^7\text{–}10^{10} \text{ cm}^{-3}$ TS8 and $y = 10^{10}\text{–}10^{12} \text{ cm}^{-3}$. Figure 5 shows the enhancement factor as a function of [PA] and [NH₃]. At a fixed NH₃ concentration of $10^{10} \text{ cm}^{-3}$ TS9, the enhancement factor weakly increases with PA concentration within the temperature range considered. The trend was observed to be similar at lower and higher PA and NH₃ concentrations, even when SA concentration increased to reach $10^8 \text{ cm}^{-3}$ TS10. The only condition that could lead to significant PA enhancements was observed for $[NH_3] > 10^{11} \text{ cm}^{-3}$ and 238 K. Figure 5 shows that for $10^{11} \text{ cm}^{-3} < [NH_3] < 10^{12} \text{ cm}^{-3}$ TS11, the enhancement factor was much more pronounced, ranging between $4.7 \times 10^2$ and $1.15 \times 10^3$. Though this enhancement factor is much lower at warmer temperatures, it is obvious that PA would actively promote sulfuric-acid-based particle formation in NH₃-polluted and cold environments. This is as expected since the promotion of new particle formation at cold temperatures has previously been evidenced (Lu et al., 2020; Liu et al., 2021). The implication of PA as participating agent in aerosol formation models would definitely reduce the errors existing in current aerosol models.

### 3.3.3 Cluster formation pathways

Gibbs free energies in Fig. 3 give information on whether the cluster formation is thermodynamically favorable at the reference pressure (1 atm); however, the concentrations of the clustering species participating in the process are not taken into account. The actual molecular clustering at given va-

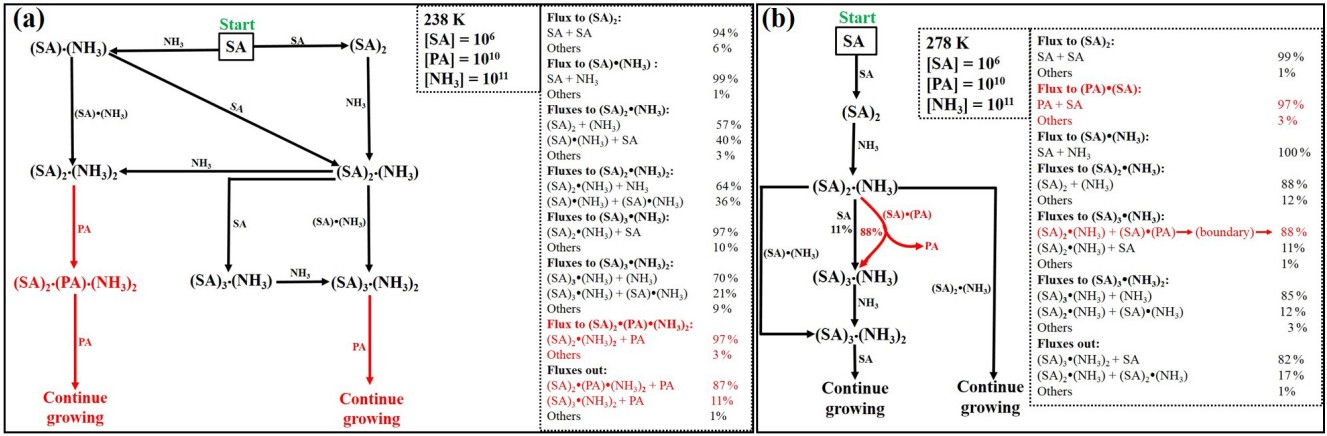

**Figure 6.** Main cluster pathways at **(a)** 238 and **(b)** 278 K. Monomer concentrations are $[SA] = 10^6 \, cm^{-3}$, $[PA] = 10^{10} \, cm^{-3}$ and $[NH_3] = 10^{11} \, cm^{-3}$ TS12. Red color indicates the path involving PA.

por concentrations and temperatures can be determined by performing ACDC simulations (McGrath et al., 2012; Olenius et al., 2013b). This is achieved by calculating the actual Gibbs free energies, that is, the vapor concentration-dependent Gibbs free energies of cluster formation, that are used to track the actual cluster formation pathways at given conditions. The temperatures considered in this study are 238, 258 and 278 K, which span most regions of the troposphere. Monomer concentrations were chosen to be $[SA] = 10^6 \, cm^{-3}$, $[PA] = 10^{10} \, cm^{-3}$, and $[NH_3] = 10^{11}$ TS13 and $5 \times 10^{11} \, cm^{-3}$ (Eger et al., 2020; Nair and Yu, 2020; Yao and Zhang, 2019; Zhang et al., 2021). As our simulation box size was set to $3 \times 2$, only clusters containing more than three acid molecules (SA and/or PA) with more than two NH₃ molecules were allowed to grow out of the system and contribute to particle formation.

Clusters containing more than one PA molecule were found not to contribute to particle formation. While the cluster contribution to the growth is found to weakly depend on monomer concentrations, their temperature dependency is relatively stronger. Depending on the temperature, the clusters grew through the system via two main paths: one path involving pure SA-NH₃ clusters and another one where PA also participates (see Figs. 6 and S2 in the Supplement). Cluster formation starts by SA collision with SA or NH₃, forming $(SA)_2$ or $SA \cdot NH_3$, followed by NH₃ or SA addition to form $(SA)_2 \cdot (NH_3)$. While PA can participate in the cluster formation as either a $(PA)_2$ or $SA \cdot PA$ cluster, it can effectively contribute to the cluster growth at low temperature (238 K), exclusively. At higher temperatures (258 and 278 K), however, PA mainly evaporates from the clusters. At these temperatures, only $(SA)_3 \cdot (NH_3)_2$ clusters will grow out of the system by clustering with SA and contribute to particle growth (See Figs. 6b and S2). At 238 K, the largest pure SA-NH₃ clusters formed within the system are $(SA)_2 \cdot (NH_3)_2$ and $(SA)_3 \cdot (NH_3)_2$, and they grow

out the system by consecutive uptake of PA monomers. The main interactions that contribute to particle formation with the participation of PA are $(SA)_3 \cdot (NH_3)_2 + PA$ and $(SA)_2 \cdot (PA)(NH_3)_2 + PA$ (see Fig. 6a).

We observed that with $[SA] = 10^6 \, cm^{-3}$, $[PA] = 10^{10} \, cm^{-3}$, and $[NH_3] = 10^{11} \, cm^{-3}$, PA-containing clusters do not contribute to the particle formation at 258 and 278 K, whereas they predominantly contribute to the particle formation at 238 K, with $\sim 100\%$ of clusters growing out of the simulation box. It follows that under the investigated monomer concentration conditions, PA-containing clusters would completely dominate the particle formation at cold temperatures while pure SA-NH₃ clusters will dominate at high temperatures.

## 4 Conclusions

The catalytic effect of pyruvic acid (PA) on SO₃ hydrolysis to form sulfuric acid (SA) and its possible enhancement in atmospheric particle formation have been highlighted. Using quantum chemical calculations, we found that with PA as a catalyst, the SO₃ hydrolysis occurs with a negative Gibbs free-energy barrier at 298 K and 1 atm, indicating a thermodynamically driven transformation process. Evaluation of the kinetics show that the rate constant of PA-catalyzed SO₃ hydrolysis at 298 K is $\sim 3 \times 10^5$ times higher than that of water-catalyzed SO₃ hydrolysis and $10^1$–$10^4$ times higher than those of previously investigated SO₃ hydrolysis processes with nitric acid, sulfuric acid, oxalic acid and formic acid acting as catalysts, hence highlighting the effective role of PA in the atmospheric chemistry of SO₃. Overall, the determination of the reaction rates, taking into account the catalyst concentrations, indicates that water-catalyzed SO₃ hydrolysis would be the main SO₃ loss pathway to form sulfuric acid under humid conditions and clean areas, whereas PA-

catalyzed $SO_3$ hydrolysis would dominate the process under dry conditions and in polluted areas.

As the PA-catalyzed $SO_3$ hydrolysis is highly stabilized by the formation of the SA·PA cluster, we further investigated the role of PA in the formation of SA-based molecular clusters. Using the quantum chemical data from our calculations and an Atmospheric Cluster Dynamics Code, we found that though PA is a weaker clustering agent to SA than SA itself, it effectively contributes to particle formation. Two main pathways were found to drive the cluster formation (one path forming pure SA-NH₃ clusters and another one forming PA-containing clusters). Under $[SA] = 10^6\,cm^{-3}$, $[PA] = 10^{10}\,cm^{-3}$ and $[NH_3] = 10^{11}\,cm^{-3}$ TS14 conditions, clusters containing more than one PA molecule were observed not to contribute to particle formation, and the main PA-containing cluster to contribute to particle formation is $(SA)_2 \cdot PA \cdot (NH_3)_2$ with the highest enhancement effect, $4.7 \times 10^2$, observed at 238 K and $[NH_3] = 1.3 \times 10^{11}\,molec.\,cm^{-3}$. This indicates that PA may actively participate in aerosol formation, especially in cold regions of the troposphere and highly NH₃-polluted environments, and may readily be included in aerosol models.

**Data availability.** All data from this research can be obtained upon request by contacting the corresponding author.

**Supplement.** The supplement related to this article is available online at: https://doi.org/10.5194/acp-22-1-2022-supplement.

**Author contributions.** NTT designed the work and performed all quantum chemical calculations. LL performed the dynamic simulations. NTT, LD, LL and XZ analyzed the data. NTT wrote the paper. LD, LL and XZ reviewed and edited the paper.

**Competing interests.** The contact author has declared that neither they nor their co-authors have any competing interests.

**Disclaimer.** Publisher's note: Copernicus Publications remains neutral with regard to jurisdictional claims in published maps and institutional affiliations.

**Acknowledgements.** We acknowledge the Wuxi Hengding Supercomputing Center Co., LTD for providing the computational resources.

**Financial support.** This work was supported by the National Natural Science Foundation of China (grant no. 21876098), Shandong Society for Environmental Science (grant no. 202001), Youth Innovation Program of Universities in Shandong Province (grant no. 2019KJD007), and Fundamental Research Fund of Shandong University (grant no. 2020QNQT012).

**Review statement.** This paper was edited by Timothy Bertram and reviewed by two anonymous referees.

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

## Remarks from the language copy-editor

## Remarks from the typesetter