# Peer review of "Pyruvic acid, an efficient catalyst in SO3 hydrolysis and effective clustering agent in sulfuric acid-based new particle formation"

_Atmospheric Chemistry and Physics, 2021_

## Author Comment (AC1)

**Reply to Comment on acp-2021-784**
Anonymous Referee #1

We thank the Referee for their constructive comments. Here, we present a point-to-point response to all the comments. For clarity, the Referee's comments are reproduced in blue color text, authors' reply are in black color and modifications to the manuscript are in red color text.

Tsona and co-workers report their results of a theoretical investigations aiming the study the role of pyruvic acid (PA) in the atmospheric formation of sulfuric acid along with the formation of clusters between pyruvic acid, sulfuric acid and ammonia, as precursors of aerosols, an issue which is interesting for atmospheric purposes. This article has two parts. The first one dealing with the role of pyruvic acid as catalyst in the hydrolysis of SO3, and the second one focused on the thermodynamic and the formation paths of the clusters.   In my opinion, the issue has potential for publication in Atmospheric Chemistry and Physics. However, I have several points to address the authors.

Regarding the role of pyruvic acid as catalyst in the hydrolysis of SO3.
1) As a general point, the authors take as zero of energies the energy of the separate species, namely SO2 + H2O + PA (or SO3 + 2H2O). Although numerically this correct, in my opinion it may not be clear at all chemically, as one would infer three particle collisions along the processes.

Though we consider three species in the beginning of the reaction, it is obvious that only two species can collide to form a binary complex that collides thereafter with the third species to form the ternary complex, which is the pre-reactive intermediate. For example, considering the water-catalyzed reaction, it would proceed through $SO_3\cdots H_2O + H_2O$ and/or $H_2O\cdots H_2O + SO_3$ interactions and form the same complex, $SO_3\cdots H_2O\cdots H_2O$. This is apparent from Fig.1 where the two pathways were clearly identified to form the same complexes, according to our calculations.
The same applies to the PA-catalyzed reaction, where $PA\cdots H_2O + SO_3$ and $PA + H_2O\cdots SO_3$ were the main identified pathways leading to the formation of the pre-reactive intermediate, $PA\cdots SO_3\cdots H_2O$.

Though this was already stated in the manuscript, we added the following text in the revised manuscript for further support:
Line 179
Indeed, due to the difficulty of termolecular interactions relative to biomolecular interaction (Buszek et al., 2012), the most likely situation is the interaction of two species to form a two-body complex followed by the interaction with the third species to form the pre-reactive intermediate.

2) In Figure 1 the authors report their results on the gas phase water hydrolysis of SO3, while the relative energies are collected in the supplementary information. Although the authors mention these previous works regarding this issue, no comparison has been done with results from the literature.

More discussion on the results of Figure 1 has been included in the revised manuscript
Line 184:

These energies are respectively within 0.17-1.81 and 0.47-1.06 kcal mol$^{-1}$ similar to previously reported values for the same reaction (Hazra and Sinha, 2011; Long et al., 2013; Torrent-Sucarrat et al., 2012; Lv et al., 2019). The slight observed differences likely result from the differences in the computational approaches used in these studies.

3) Regarding the reaction mechanism, there are two ways in which the hydrolysis can take place. The first one, involve the reaction of SO3···H2O with PA and the second one the reaction of PA···H2O + SO3. Along the text the authors suggest that preferably proceeds by reaction of PA···H2O + SO3. From Figure 2, it seems that the authors consider only the reaction of SO3··H2O with the most stable PA (PAtc) whereas the reaction of the naked SO3 takes place with the PAtl and Pact complexes. In my opinion the authors miss some preliminary steps in the reaction mechanisms, which may have effect on the kinetics.
   a) The SO3···H2O react with both PAtl and Pact so that these processes should be considered.

According to our calculations, $PA_{Tc}$ could interact with $H_2O$ and be converted into $PA_{Tt}$, forming the $PA_{Tt}\cdots H_2O$ binary complex. At the same time, $PA_{Tt}$ interacts with $H_2O$ to form the $PA_{Tt}\cdots H_2O$ binary complex. Further interactions, $PA_{Tt}\cdots H_2O + SO_3$ and $PA_{Tt} + SO_3\cdots H_2O$, both lead to the formation of the same $PA_{Tt}\cdots H_2O\cdots SO_3$ pre-reactive intermediate. Likewise, $PA_{Ct}\cdots H_2O + SO_3$ and $PA_{Ct} + SO_3\cdots H_2O$ interactions lead to the same $PA_{Ct}\cdots H_2O\cdots SO_3$ pre-reactive intermediate. Hence, the following equilibria are involved in the formation of pre-reactive intermediates:

$$SO_3 + H_2O \leftrightarrow SO_3\cdots H_2O, K_{eq1,SO3} \tag{R1}$$
$$PA_{Tt} + H_2O \leftrightarrow PA_{Tt}\cdots H_2O, K_{eq1,PATt} \tag{R2}$$
$$PA_{Ct} + H_2O \leftrightarrow PA_{Ct}\cdots H_2O, K_{eq1,PACt} \tag{R3}$$
$$PA_{Tt}\cdots H_2O + SO_3 \leftrightarrow PA_{Tt}\cdots H_2O\cdots SO_3, K_{eq2,RC\text{-}PATt,a} \tag{R4a}$$
$$SO_3\cdots H_2O + PA_{Tt} \leftrightarrow PA_{Tt}\cdots H_2O\cdots SO_3, K_{eq2,RC\text{-}PATt,b} \tag{R4b}$$
$$PA_{Ct}\cdots H_2O + SO_3 \leftrightarrow PA_{Ct}\cdots H_2O\cdots SO_3, K_{eq2,RC\text{-}PACt,a} \tag{R5a}$$
$$SO_3\cdots H_2O + PA_{Ct} \leftrightarrow PA_{Ct}\cdots H_2O\cdots SO_3, K_{eq2,RC\text{-}PACt,b} \tag{R5b}$$

where $K_{eq}$ are corresponding equilibrium constants. Although in each case the pre-reactive intermediate is formed from two different interactions, we chose to include only the most energetically favorable one in **Fig. 2** for the sake of clarity. Nevertheless, the contribution of each path to the formation of the pre-reactive intermediate can be quantified by determining the relative concentrations of $SO_3\cdots H_2O$ and $PA\cdots H_2O$ that actively participate in the formation of the pre-reactive intermediate. Application of the law of mass action to Eqs. (R1)-(R5) leads to the following expressions for $PA\cdots H_2O$ to $SO_3\cdots H_2O$ ratio:

$$\frac{[PA_{Tt}\cdots H_2O]}{[SO_3\cdots H_2O]} = \frac{K_{eq,PATt}[PA_{Tt}]}{K_{eq,SO3}[SO_3]} \text{ and } \frac{[PA_{Ct}\cdots H_2O]}{[SO_3\cdots H_2O]} = \frac{K_{eq,PACt}[PA_{Ct}]}{K_{eq,SO3}[SO_3]}.$$

Considering atmospheric PA and $SO_3$ concentrations of $\sim 10^{10}$ cm$^{-3}$ and $\sim 10^6$ cm$^{-3}$, respectively, and taking into account the relative abundances of each PA conformer (0.95 for $PA_{Tc}$, 0.04 for $PA_{Tt}$ and 0.01 for $PA_{Ct}$), we find that regardless of the PA conformer, $PA\cdots H_2O$ will contribute

by more than 99.99% to the formation of PA···H₂O···SO₃ while SO₃···H₂O will contribute by less than 0.01% (details are given in the Supplement). Overall, the pathway through SO₃···H₂O + PA has no net effect on the kinetics of H₂SO₄ formation, as it nearly does not contribute to the formation the pre-reactive intermediate. The formation of PA···H₂O···SO₃ is essentially controlled by PA···H₂O + SO₃ collision, hereby supporting the sole inclusion of this pathway in **Fig. 2**.

To clarify, we added the following at Line 224:
Moreover, to evaluate the impact of each of these paths to the formation of the pre-reactive intermediate, we determined that among the two hydrates, PA···H₂O will contribute by more than 99.99% while SO₃···H₂O will contribute by less than 0.01% to the formation of the pre-reactive intermediate regardless of the PA conformer (details are given in the Supplement).

> b) The PAtl···H2O and Pact···H2O complexes are held together be two hydrogen bonds, one between the acidic hydrogen of PA and the oxygen atom of water, and the other between with the carbonyl oxygen of PA with one hydrogen atom of H2O. In both cases the complexes form a ring structure which should be broken to from the RCtl and RCct complexes. This requires the existence of a transition state that should be taken into account. Moreover, for instance, the pre-reactive complexes PA···H2O··SO3 (namely RC's) arising from PA···H2O + SO3 could also decompose, without energy barrier, into PA + SO3···H2O which should be considered.

It is obvious that the approaches of SO₃ and PA towards PA···H₂O and SO₃···H₂O, respectively, lead to rearrangements prior to the formation of the pre-reactive intermediate, PA···H₂O···SO₃. However, these rearrangements are not believed to lead to the formation of other particular minima prior to PA···H₂O···SO₃ formation. DePalma et al. demonstrated that minima formed from such rearrangements would be separated by transition states commonly larger than $RT$ (DePalma et al., 2014). Furthermore, following the energy paths in SO₃ + PA···H₂O and PA + SO₃···H₂O optimizations from our calculations indicates that the formation of PA···H₂O···SO₃ was downhill, being ~13 to 15 kcal mol⁻¹ exothermic, with no remarkable barrier. For such low-lying complex formation, the decomposition of the pre-reactive intermediate to PA + SO₃···H₂O or SO₃ + PA···H₂O would be highly prevented, relative to its conversion to PA···H₂SO₄ by overcoming energy barriers around 0 kcal mol⁻¹ or below. Hence, the kinetics of PA···H₂O···SO₃ formation either from SO₃ + PA···H₂O or PA + SO₃···H₂O would likely not be affected by other intermediate steps, if not very weakly, while its decomposition is mainly towards PA···H₂SO₄ formation, its most likely fate. All fundamental steps controlling PA···H₂O···SO₃ sources and sinks are depicted by reaction (R2) in the main manuscript, assuming pseudo steady-state approximation, and they are accounted for in the overall kinetics through their equilibrium constants and the rate constant of unimolecular decomposition of PA···H₂O···SO₃, executed by the KiSThelP program.

> 4) It is not clear how the authors have considered all these reactions weighted according the concentration of the different reactants (naked or forming complexes with water). The authors do not report values of the kinetic/equilibrium constants (for instance for the rection of the reactant complexes) at different temperatures to support the discussion in the

paragraph 215. This information is necessary and the values of the bimolecular rate constants should be detailed. By the same way, the discussion in paragraph 240 should be supported by the specific values of concentrations and relative humidity.

Considering that different PA conformers would coexist in the atmosphere, we determined their equilibrium distribution at 298 K in order to take into account the effective role of PA in the kinetics. Using the Gibbs free energy changes of the equilibria below

$PA_{Tc} \leftrightarrow PA_{Tt}$

$PA_{Tc} \leftrightarrow PA_{Ct}$

$PA_{Tc} \leftrightarrow PA_{Cc}$

and applying the law of mass action, we determined the relative abundances of PA conformers at 298 K and their values are given in the **Table R1** below, along with their formation Gibbs free energies relative to the energy of $PA_{Tc}$.

**Table R1** Gibbs free energies ($\triangle$G) values of PA conformers, calculated at the M06-2X/6-311++G(3df,3pd) level of theory relative to $PA_{Tc}$ energy, and their corresponding relative abundance. Energies are calculated at 298 K and 1 atm, and units are kcal mol$^{-1}$.

| Conformer | $PA_{Tc}$ | $PA_{Tt}$ | $PA_{Ct}$ | $PA_{Cc}$ |
|---|---|---|---|---|
| $\triangle$G | 0 | 1.81 | 3.13 | 9.18 |
| Relative abundance | 0.95 | 0.04 | 0.01 | 0.00 |

Following the expression for the reaction rate given by **Eq. (1)** in the main manuscript, the numerical values of equilibrium constants and rates constants were calculated by the following equations.

The rate constant of the unimolecular decomposition of the pre-reactive intermediate was calculated as:

$$K_{uni} = \Gamma \times \frac{k_B T}{h c^0} \times \exp\left(-\frac{\Delta G^{\#}}{RT}\right) \qquad (1)$$

where $\Delta G^{\#}$ is the Gibbs free energy barrier separating the pre-reactive intermediate and the products, $h$ is the Planck's constant, $R$ is the molar gas constant, $c^0$ is the standard gas-phase concentration, and $\Gamma$ is the Wigner tunneling correction.

For two species $A$ and $B$ interacting to form a third species $C$ ($A + B \leftrightarrow C$), the equilibrium constant is calculated as:

$$K_{eq} = \exp\left(-\frac{\Delta G}{RT}\right) \qquad (2)$$

where $\triangle$G is the change in standard Gibbs free energy.

The values of equilibrium constants for $SO_3$ and PA hydration, rate constants of unimolecular decomposition of the pre-reactive intermediates and rate constants of $H_2O$/PA-catalyzed $SO_3$ hydrolysis, calculated at 298 K are shown in **Tables S2** and **S3** in the Supplement.

The sentence at Line 123 was modified as given below to highlight the numerical values of equilibrium constants and reaction rate constants:

The determination of equilibrium constants and unimolecular rate constants was executed using the KiSThelP program, and their numerical values are given in **Tables S2** and **S3** in the Supplement.

5) Regarding the cluster formation:
a) My main point here is why the authors do not have included any water molecule in the formation of clusters. It is well known that sulfuric acid is fully hydrated at ground level in the atmosphere (see references in line 490 and ff among others) so that water should play a role in the formation of these clusters. This issue must be discussed.

Despite the well-known clustering ability of water to sulfuric acid, several theoretical and experimental studies have shown that water would rather evaporate back from the cluster, owing to its high evaporation rate, more especially as the number of sulfuric acid molecules in the cluster increases and when there are base molecules present.

For clarifications, we have added the following in the revised manuscript.

Line 126:

Although sulfuric acid as a monomer is known to cluster effectively with water at relevant atmospheric conditions, several studies have demonstrated that the binding strength of water significantly decreases as sulfuric acid is clustered to other sulfuric acid and base molecules (Temelso et al., 2012a; Temelso et al., 2012b; Henschel et al., 2014; Tsona et al., 2015). Moreover, clusters dynamics simulations have shown that despite its clustering to sulfuric acid and sulfuric acid-based clusters, water rapidly evaporates from the cluster owing to its high evaporation rate, more especially when base molecules are present in the cluster. As a result, most atmospheric sulfuric acid-based clusters are detected in their dry state, exclusively (Almeida et al., 2013; Olenius et al., 2013). It follows that though water would play a certain role in the cluster thermodynamics, it net effect on particle formation rate is negligible. Hence, water was not included in the studied clusters.

b) Another point is how the authors have chosen the structures to calculate the clusters. Have they performed a previous scan or a dynamic calculation to select them?

To select the cluster structures, several initial configurations were generated manually by arranging the participating molecules in different directions and pre-optimizing them. By step-wise addition of monomers to a cluster, larger clusters were built. Depending on the cluster size, 10-30 initial configurations of each cluster were pre-optimized at the M062X/6-31+G(d) level of theory and the best structures with energies within 3 kcal mol$^{-1}$ similar to the lowest energy configuration were re-optimized at the M062X/6-31++G(d,p) level of theory. Although the 6-31++G(d,p) basis set is somewhat smaller than the 6-311++G(3df,3pd) basis set used in the thermodynamics and reaction kinetics part, a benchmark study has shown that the use of the 6-31++G(d,p) basis set instead of 6-311++G(3df,3pd) in modeling sulfuric acid-based clusters only introduces low errors in the Gibbs free energy, yet significantly reducing the computation cost (Elm and Mikkelsen, 2014). Hence, we chose the M062X/6-31++G(d,p) method for optimizing the modeled clusters.

The following was added in the revised manuscript for clarifications:

Line 135:

A number of initial configurations of SA-NH$_3$ clusters were taken from previously published results (Ortega et al., 2012) and re-optimized with the M06-2X/6-31++G(d,p), while those containing PA were built by stepwise addition of monomers to the relevant cluster. On this basis, several starting configurations were generated manually by arranging the participating

molecules/clusters in different directions. Depending on the cluster size, 10-30 initial configurations of each cluster were pre-optimized at the M062X/6-31+G(d) level of theory and all identified structures within 3 kcal mol$^{-1}$ of the lowest energy structure were thereafter re-optimized with the M06-2X/6-31++G(d,p) method and the vibrational frequency analysis were subsequently performed at the same level of theory. It has been shown that the reduction from 6-311++G(3df,3pd) to 6-31++G(d,p) basis set for sulfuric acid-based cluster formation induces very little errors in the thermal contribution to the Gibbs free energy, with no further substantial effect on the single point energy, yet sufficiently reducing the computation cost (Elm and Mikkelsen, 2014). The electronic energies of M06-2X/6-31++G(d,p) optimized structures were further corrected with the DLPNO-CCSD(T)/aug-cc-pVTZ method.

**References**

Almeida, J., Schobesberger, S., Kuerten, A., Ortega, I. K., Kupiainen-Maatta, O., Praplan, A. P., Adamov, A., Amorim, A., Bianchi, F., Breitenlechner, M., David, A., Dommen, J., Donahue, N. M., Downard, A., Dunne, E., Duplissy, J., Ehrhart, S., Flagan, R. C., Franchin, A., Guida, R., Hakala, J., Hansel, A., Heinritzi, M., Henschel, H., Jokinen, T., Junninen, H., Kajos, M., Kangasluoma, J., Keskinen, H., Kupc, A., Kurten, T., Kvashin, A. N., Laaksonen, A., Lehtipalo, K., Leiminger, M., Leppa, J., Loukonen, V., Makhmutov, V., Mathot, S., McGrath, M. J., Nieminen, T., Olenius, T., Onnela, A., Petaja, T., Riccobono, F., Riipinen, I., Rissanen, M., Rondo, L., Ruuskanen, T., Santos, F. D., Sarnela, N., Schallhart, S., Schnitzhofer, R., Seinfeld, J. H., Simon, M., Sipila, M., Stozhkov, Y., Stratmann, F., Tome, A., Troestl, J., Tsagkogeorgas, G., Vaattovaara, P., Viisanen, Y., Virtanen, A., Vrtala, A., Wagner, P. E., Weingartner, E., Wex, H., Williamson, C., Wimmer, D., Ye, P., Yli-Juuti, T., Carslaw, K. S., Kulmala, M., Curtius, J., Baltensperger, U., Worsnop, D. R., Vehkamaki, H., and Kirkby, J.: Molecular understanding of sulphuric acid-amine particle nucleation in the atmosphere, Nature, 502, 359-+, 10.1038/nature12663, 2013.

Buszek, R. J., Barker, J. R., and Francisco, J. S.: Water Effect on the OH plus HCl Reaction, Journal of Physical Chemistry A, 116, 4712-4719, 10.1021/jp3025107, 2012.

DePalma, J. W., Bzdek, B. R., Ridge, D. P., and Johnston, M. V.: Activation Barriers in the Growth of Molecular Clusters Derived from Sulfuric Acid and Ammonia, The Journal of Physical Chemistry A, 118, 11547-11554, 10.1021/jp507769b, 2014.

Elm, J., and Mikkelsen, K. V.: Computational approaches for efficiently modelling of small atmospheric clusters, Chemical Physics Letters, 615, 26-29, https://doi.org/10.1016/j.cplett.2014.09.060, 2014.

Hazra, M. K., and Sinha, A.: Formic acid catalyzed hydrolysis of SO3 in the gas phase: a barrierless mechanism for sulfuric acid production of potential atmospheric importance, J Am Chem Soc, 133, 17444-17453, 10.1021/ja207393v, 2011.

Henschel, H., Navarro, J. C. A., Yli-Juuti, T., Kupiainen-Määttä, O., Olenius, T., Ortega, I. K., Clegg, S. L., Kurtén, T., Riipinen, I., and Vehkamäki, H.: Hydration of Atmospherically Relevant Molecular Clusters: Computational Chemistry and Classical Thermodynamics, The Journal of Physical Chemistry A, 118, 2599-2611, 10.1021/jp500712y, 2014.

Long, B., Chang, C.-R., Long, Z.-W., Wang, Y.-B., Tan, X.-F., and Zhang, W.-J.: Nitric acid catalyzed hydrolysis of SO3 in the formation of sulfuric acid: A theoretical study, Chemical Physics Letters, 581, 26-29, https://doi.org/10.1016/j.cplett.2013.07.012, 2013.

Lv, G., Sun, X., Zhang, C., and Li, M.: Understanding the catalytic role of oxalic acid in SO3 hydration to form H2SO4 in the atmosphere, Atmos. Chem. Phys., 19, 2833-2844, 10.5194/acp-19-2833-2019, 2019.

Olenius, T., Schobesberger, S., Kupiainen-Määttä, O., Franchin, A., Junninen, H., Ortega, I. K., Kurtén, T., Loukonen, V., Worsnop, D. R., Kulmala, M., and Vehkamäki, H.: Comparing simulated and experimental molecular cluster distributions, Faraday Discussions, 165, 75-89, 10.1039/C3FD00031A, 2013.

Ortega, I. K., Kupiainen, O., Kurtén, T., Olenius, T., Wilkman, O., McGrath, M. J., Loukonen, V., and Vehkamäki, H.: From quantum chemical formation free energies to evaporation rates, Atmos. Chem. Phys., 12, 225-235, 10.5194/acp-12-225-2012, 2012.

Temelso, B., Morrell, T. E., Shields, R. M., Allodi, M. A., Wood, E. K., Kirschner, K. N., Castonguay, T. C., Archer, K. A., and Shields, G. C.: Quantum Mechanical Study of Sulfuric Acid Hydration: Atmospheric Implications, The Journal of Physical Chemistry A, 116, 2209-2224, 10.1021/jp2119026, 2012a.

Temelso, B., Phan, T. N., and Shields, G. C.: Computational Study of the Hydration of Sulfuric Acid Dimers: Implications for Acid Dissociation and Aerosol Formation, The Journal of Physical Chemistry A, 116, 9745-9758, 10.1021/jp3054394, 2012b.

Torrent-Sucarrat, M., Francisco, J. S., and Anglada, J. M.: Sulfuric acid as autocatalyst in the formation of sulfuric acid, J Am Chem Soc, 134, 20632-20644, 10.1021/ja307523b, 2012.

Tsona, N. T., Henschel, H., Bork, N., Loukonen, V., and Vehkamäki, H.: Structures, Hydration, and Electrical Mobilities of Bisulfate Ion–Sulfuric Acid–Ammonia/Dimethylamine Clusters: A Computational Study, The Journal of Physical Chemistry A, 119, 9670-9679, 2015.

---

## Author Comment (AC2)

**Reply to Comment on acp-2021-784**
Anonymous Referee #2

We thank the Referee for their constructive comments. Here, we present a point-to-point response to all the comments. For clarity, the Referee's comments are reproduced in blue color text, authors' reply are in black color and modifications to the manuscript are in red color text.

The manuscript titled "Pyruvic acid, an efficient catalyst in SO3 hydrolysis and effective clustering agent in sulfuric acid-based new particle formation" by Tsona et al. discusses the role of pyruvic acid as a catalyst in SO3 hydrolysis. The investigation of whether pyruvic acid can take part in cluster formation brings an interesting addition to the article. The paper is well written and it fits to the scope of the journal.

Specific comments:
1. I would like to see a section in the Methodology on how the cluster conformers were found for the ACDC calculations. Presumably the clusters containing only sulfuric acid and ammonia can be found somewhere in the literature, but how about the clusters containing pyruvic acid? I can see that for the hydrolysis calculations, different conformers of the pyruvic acid were used, but the clusters should have a lot of conformers due to the higher number of molecules as well as the 4 different pyruvic acid conformers, which should be considered in finding the lowest energy cluster.

To select the cluster structures, several initial configurations were generated manually by arranging the participating molecules in different directions and pre-optimizing them. By step-wise addition of monomer to a cluster, larger clusters were built. Depending on the cluster size, 10-30 initial configurations of each cluster were pre-optimized at the M062X/6-31+G(d) level of theory and the best structures with energies within 3 kcal mol$^{-1}$ similar to the lowest energy configuration were re-optimized at the M062X/6-31++G(d,p) level of theory. Although the 6-31++G(d,p) basis set is somewhat smaller than the 6-311++G(3df,3pd) basis set used in the thermodynamics and reaction kinetics part, a benchmark study has shown that the use of the 6-31++G(d,p) basis set instead of 6-311++G(3df,3pd) in modeling sulfuric acid-based clusters only introduces low errors in the Gibbs free energy, yet significantly reducing the computation cost (Elm and Mikkelsen, 2014). Hence, we chose the M062X/6-31++G(d,p) method for optimizing the modeled clusters.
The following was added in the revised manuscript for clarifications:
Line 135:
A number of initial configurations of SA-NH$_3$ clusters were taken from previously published results (Ortega et al., 2012) and re-optimized with the M06-2X/6-31++G(d,p), while those containing PA were built by stepwise addition of monomers to the relevant cluster. On this basis, several starting configurations were generated manually by arranging the participating molecules/clusters in different directions. Depending on the cluster size, 10-30 initial configurations of each cluster were pre-optimized at the M062X/6-31+G(d) level of theory and all identified structures within 3 kcal mol$^{-1}$ of the lowest energy structure were thereafter re-optimized with the M06-2X/6-31++G(d,p) method and the vibrational frequency analysis were

subsequently performed at the same level of theory. It has been shown that the reduction from 6-311++G(3df,3pd) to 6-31++G(d,p) basis set for sulfuric acid-based cluster formation induces very little errors in the thermal contribution to the Gibbs free energy, with no further substantial effect on the single point energy, yet sufficiently reducing the computation cost (Elm and Mikkelsen, 2014). The electronic energies of M06-2X/6-31++G(d,p) optimized structures were further corrected with the DLPNO-CCSD(T)/aug-cc-pVTZ method.

2. In Fig. 5 (bottom panel), can you discuss why are the enhancement factors of 238 K at 3 and 4*10^9 PA molecules/cm-3 the same, when all other points seem to have the same linear trend with the same slope?

It should be noted that the enhancement factor is calculated as the ratio of the particle formation rate in the SA-PA-NH$_3$ system to that in the SA-NH$_3$ system as shown by **Eq. (6)** in the main manuscript. At [SA] = $10^6$ cm$^{-3}$ and [NH$_3$] = $10^{10}$ cm$^{-3}$, with [PA] increasing from $10^9$ to $10^{10}$ cm$^{-3}$, the increase in particle formation rate is very subtle, with slight differences appearing only from the fifth decimal. As a consequence, the change in the enhancement factor as [PA] increases is also very weak. The seemingly similar PA enhancement factor at [PA] = $3\times10^9$ cm$^{-3}$ and [PA] = $4\times10^9$ cm$^{-3}$ is due to the way the data were truncated, being at the third decimal. More accurate values of the enhancement factor can be obtained by truncating the values at fourth decimal. The revised plot is shown below and it has been uploaded in the revised manuscript.

[Figure]

**Figure 5: Enhancement of PA in the clusters formation rate in the sulfuric acid-pyruvic acid-ammonia clusters at [SA] = $10^6$ cm$^{-3}$, [NH$_3$] = $10^{10}$ cm$^{-3}$, [PA] = $10^9$ -$10^{10}$ cm$^{-3}$ and different temperatures (bottom panel), and T = 238 K, [SA] = $10^6$ cm$^{-3}$, [PA] = $10^{10}$ cm$^{-3}$, [NH$_3$] = $10^{11}$ -$10^{12}$ cm$^{-3}$ (top panel).**

3. Did you run any simulation wher you would have an initial concentration of PASA, formed during the SA formation? Would this have any effect on the cluster formation results or would the PA just evaporate from the cluster, once more SA and NH3 is added through collisions?
We did not simulate the SA•PA concentration from the PA-catalyzed $SO_3$ hydrolysis, since this would combine not only the kinetics but also dynamics of all involved species, including water for which the concentration is 7 to 10 orders of magnitude higher than the concentrations of other species ($SO_3$ and PA). Due to this high concentration difference, the kinetic modeling of the studied system would be impossible, as the collision frequency of water with other species would be 7-10 orders of magnitude higher than that of $SO_3$ of PA, resulting in extremely stiff set of equations that cannot be solved practically (Paasonen et al., 2012).

Instead, the SA•PA concentration was determined in the cluster dynamics simulations by solving the birth-death equations at given monomer concentrations. Fig. 4 indicates that SA•PA concentration would rarely exceed $10^4$ $cm^{-3}$ at most relevant monomer concentrations. Our dynamics simulations further indicated that though PA forms clusters with SA in the system, it would rapidly evaporate back unless the cluster has reached a certain size and at low temperatures, exclusively. This was discussed in Section 3.3.2.

Technical corrections:
line 21: "The enhancing effect of PA of examined by evaluating the ratio of the ternary..." There is some typo here.
This has been corrected to "The enhancing effect of PA examined by evaluating the ratio of the ternary…"

line 55, 222, 245: giving -> given
Corrections have been made at the indicated places

line 58: acid in the troposphere -> acids in the troposphere
This has been corrected

line 85: You mention only zero-point energies, though later you use also Gibbs free energies. Did you get them also from this calculation?

Both thermal corrections to the Gibbs free energy and the zero-point energies were calculated in the current study, using the same method. Throughout, we use both zero-point corrected electronic energies and Gibbs free energies to describe the energetics and thermodynamics of the studied systems.

In the revised manuscript, the sentence at Line 85 was modified to highlight the thermal contribution to the Gibbs free energy as follows:

Identified M06-2X/6-31+G(d,p) structures within 3 kcal $mol^{-1}$ of the lowest energy structure were re-optimized, followed by vibrational frequencies analysis at the M06-2X/6-311++G(3df,3pd) level of theory, thereby yielding zero-point energies as well as thermal correction to Gibbs free energies.

line 88: "internal reaction coordinate" should be "intrinsic reaction coordinate" for IRC.
This has been corrected

line 134: Is there a typo in the birth-death equation, the concentration term is missing from end (the cluster evaporation sink term).
The equation has been revised as follows:

$$\frac{dc_i}{dt} = \frac{1}{2}\sum_{j<i}\beta_{j,(i-j)}C_jC_{i-j} + \sum_j \gamma_{(i+j)\to i,j}C_{i+j} - \sum_j \beta_{i,j}C_iC_j - \frac{1}{2}\sum_{j<i}\gamma_{i\to j,(i-j)} + Q_i - S_i$$

(2)

where $C_i$ is the concentration of cluster $i$, $\beta_{i,j}$ is the collision coefficient of clusters $i$ and $j$, $\gamma_{k\to i,j}$ is the rate coefficient of cluster $k$ evaporating into smaller clusters $i$ and $j$. $Q_i$ and $S_i$ are possible outside source term and sink term, respectively, for cluster $i$.

line 156: "second molecule" -> "second H2O molecule"
This has been corrected.

**References**

Elm, J., and Mikkelsen, K. V.: Computational approaches for efficiently modelling of small atmospheric clusters, Chemical Physics Letters, 615, 26-29, https://doi.org/10.1016/j.cplett.2014.09.060, 2014.

Ortega, I. K., Kupiainen, O., Kurtén, T., Olenius, T., Wilkman, O., McGrath, M. J., Loukonen, V., and Vehkamäki, H.: From quantum chemical formation free energies to evaporation rates, Atmos. Chem. Phys., 12, 225-235, 10.5194/acp-12-225-2012, 2012.

Paasonen, P., Olenius, T., Kupiainen, O., Kurtén, T., Petäjä, T., Birmili, W., Hamed, A., Hu, M., Huey, L. G., Plass-Duelmer, C., Smith, J. N., Wiedensohler, A., Loukonen, V., McGrath, M. J., Ortega, I. K., Laaksonen, A., Vehkamäki, H., Kerminen, V. M., and Kulmala, M.: On the formation of sulphuric acid – amine clusters in varying atmospheric conditions and its influence on atmospheric new particle formation, Atmos. Chem. Phys., 12, 9113-9133, 10.5194/acp-12-9113-2012, 2012.